# Application of Supercritical CO_2_ Extraction and Identification of Polyphenolic Compounds in Three Species of Wild Rose from Kamchatka: *Rosa acicularis*, *Rosa amblyotis*, and *Rosa rugosa*

**DOI:** 10.3390/plants14010059

**Published:** 2024-12-27

**Authors:** Mayya P. Razgonova, Muhammad A. Nawaz, Elena A. Rusakova, Kirill S. Golokhvast

**Affiliations:** 1N.I. Vavilov All-Russian Institute of Plant Genetic Resources, B. Morskaya 42-44, 190000 Saint-Petersburg, Russia; golokhvast@sfsca.ru; 2Far Eastern Federal University, Sukhanova 8, 690950 Vladivostok, Russia; 3Advanced Engineering School “Agrobiotek”, National Research Tomsk State University, Lenin Ave, 36, 634050 Tomsk, Russia; 4FSBSI Kamchatsky Scientific Research Institute of Agriculture, Centralnaya, 4, 684033 Sosnovka, Russia; rubusarcticus@mail.ru; 5Siberian Federal Scientific Centre of Agrobiotechnology RAS, Centralnaya 2b, Presidium, 633501 Krasnoobsk, Russia

**Keywords:** *Rosa acicularis*, *Rosa amblyotis*, *Rosa rugosa*, tandem mass spectrometry, SC-CO_2_ extraction, polyphenols, metabolome

## Abstract

A comparative metabolomic study of three varieties of wild Rosa (*Rosa acicularis*, *Rosa amblyotis*, and *Rosa rugosa*) from a Kamchatka expedition (2024) was conducted via extraction with supercritical carbon dioxide modified with ethanol (EtOH), and detection of bioactive compounds was realized via tandem mass spectrometry. Several experimental conditions were investigated in the pressure range 50–350 bar, with the used volume of co-solvent ethanol in the amount of 2% in the liquid phase at a temperature in the range of 31–70 °C. The most effective extraction conditions are the following: pressure 200 Bar and temperature 55 °C for *Rosa acicularis*; pressure 250 Bar and temperature 60 °C for *Rosa amblyotis*; pressure 200 Bar and temperature 60 °C for *Rosa rugosa*. Three varieties of wild *Rosa* contain various phenolic compounds and compounds of other chemical groups with valuable biological activity. Tandem mass spectrometry (HPLC-ESI–ion trap) was applied to detect the target analytes. A total of 283 bioactive compounds (two hundred seventeen compounds from the polyphenol group and sixty-six compounds from other chemical groups) were tentatively identified in extracts from berries of wild *Rosa*. For the first time, forty-eight chemical constituents from the polyphenol group (15 flavones, 14 flavonols, 4 flavan-3-ols, 3 flavanones, 1 phenylpropanoid, 2 gallotannins, 1 ellagitannin, 4 phenolic acids, 1 dihydrochalcone, and 3 coumarins) were identified in supercritical extracts of *R. acicularis*, *R. amblyotis*, and *R. rugosa*.

## 1. Introduction

The genus Rosa (family *Rosaceae*) is represented on the territory of Kamchatka by three species: *Rosa rugosa* Thumb., *Rosa amblyotis* C.A. Mey., and *Rosa acicularis* Lindl. Fresh fruits and leaves contain up to 900 mg% ascorbic acid per dry pulp weight. Fresh petals contain 0.25–0.38% essential oil. Its neutral volatile fraction contains 86.3% phenylethyl alcohol and some linalool, citronellol, geraniol, nerol, etc. Eugenol was found in the phenolic fraction, phenylacetic, benzoic, and other acids in the acid fraction. *R. rugosa* is a medicinal plant widely used in traditional and folk medicine. Extracts of *R. rugosa* have been valued for Asian culinary, cosmetic, and aromatherapy purposes and used in herbal medicines for diabetes mellitus and osteoarthritis [1]. In Korea, *R. rugosa* has been used as a traditional medicine for its anti-inflammatory, antioxidant, hypoglycemic, and hypocholesterolemic properties. These properties may be primarily associated with both the complex of polyphenolic compounds, including tannins, flavonoids, etc., and the terpene chemical group [2]. The medicinal effects seem to be involved in the presence of many phytochemicals in *R. rugosa* extracts, for example flavonoids, phenylpropanoid, tannins, fatty acids, and terpenoids [3].

Phenolic compounds are secondary plant metabolites whose presence and composition are important indicators of the nutraceutical and nutritional quality of fruits, vegetables, and other plants [4,5]. These compounds have an aromatic ring bearing one or more hydroxyl groups, and phenolic structures can range from a simple phenolic molecule to complex high-molecular polymers [6]. These compounds, as one of the most widespread groups of phytochemicals, directly affect plant physiology and morphology. Phenolic compounds can act as phytoalexins [7], pollinator attractants, antifeedants, plant pigmentation factors, and antioxidants and have a protective function against ultraviolet light [8]. A group of polyphenolic compounds directly contributes to the color and flavor characteristics of fruits and vegetables [9]. Natural polyphenolic compounds have been reported to have excellent properties as food preservatives and play a major role in protecting against several pathological disorders in the human body, such as atherosclerosis, brain dysfunction, and cancer (Gordon, 1996). Moreover, polyphenols also have many industrial applications; for example, polyphenols directly obtained from plant extracts can be used as natural colorants and preservatives for food products or in the production of paints, paper, and cosmetics [10].

Supercritical fluid extraction with the use of pressured CO_2_ (SC-CO_2_) has been used since the last 50 years in analytical methodologies to investigate the composition of food products, removal of undesirable substances, and isolation of the polyphenol group of compounds and other chemical constituents. The goal of this work was to identify and select polyphenolic compounds from three varieties of wild *Rosa* (*R. acicularis*, *R. amblyotis*, and *R. rugosa*) via extraction with SC-CO_2_. Also, a tandem mass spectrometry protocol was used for the detailed screening of phytochemicals present in three varieties of wild *Rosa*.

## 2. Materials and Methods

### 2.1. Materials

The subject of this study was the berries of wild *Rosa* (*R. acicularis*, *R. amblyotis*, and *R. rugosa*) collected and grown at the expedition work (Kamchatka) of N.I. Vavilov All-Russian Institute of Plant Genetic Resources, as depicted in Figure 1. The berries of *R. acicularis*, *R. amblyotis*, and *R. rugosa* were harvested at the end of July 2024. All plant tissues used in this work were conformed according to the standard established by the State Pharmacopoeia of the Russian Federation [11].

### 2.2. Chemicals and Reagents

All reagents used in this study were of analytical grade. HPLC-grade acetonitrile was purchased from Fisher Scientific (Southborough, UK), and MS-grade formic acid and ethanol (EtOH) were purchased from Sigma-Aldrich (Steinheim, Germany). Ultrapure water was prepared from Siemens (SIEMENS water technologies, Munich, Germany).

### 2.3. Extraction

SC-CO_2_ extraction was performed using the SFE-500 supercritical pressure extraction system (Thar SCF Waters, Milford, CT, USA). System options include the following: a co-solvent pump (Thar Waters P-50 High Pressure Pump) for the extraction of polar samples; a CO_2_ flow meter (Siemens, Germany) to measure the amount of CO_2_ supplied to the system; and multiple extraction vessels to extract different sample sizes or to increase the throughput of the system. The flow rate was 10–25 mL/min for liquid CO_2_ and 1.00 mL/min for EtOH. Extraction samples of 200 g of berries (*R. acicularis*) were used. Extraction samples of 200 g of berries (*R. amblyotis*) were used. Extraction samples of 200 g of berries (*R. rugosa*) were used. Extraction time was counted after reaching working pressure and equilibrium flow and was 60–90 min for each sample. This method of SC-CO_2_ extraction of plant matrices was tested by the authors on numerous plant samples, including aboveground and underground parts of the plant [12,13].

### 2.4. Liquid Chromatography

HPLC was performed using a Shimadzu LC-20 Prominence HPLC (Shimadzu, Kyoto, Japan) equipped with a UV sensor and C18 silica reverse phase column (4.6 × 150 mm, particle size: 2.7 μm) to perform the separation of multicomponent mixtures. The mobile phase eluent A was deionized water containing 0.1% formic acid, and eluent B was acetonitrile containing 0.1% formic acid. The gradient elution was started at 0–2 min, with 0% eluent B for 2–50 min, and 0–100% B; control washing for 50–60 min with 100% B. The mobile phase flow rate and column temperature were maintained at 0.3 mL/min and 30 °C, respectively. The entire HPLC analysis was performed with a UV–vis detector SPD-20A (Shimadzu, Kyoto, Japan) at a wavelength of 230 nm for the identification of compounds. The injection volume was 10 μL. Additionally, liquid chromatography was combined with a mass spectrometric ion trap to identify compounds.

### 2.5. Mass Spectrometry

MS analysis was performed on an ion trap amaZon SL (Bruker Daltoniks, Leipzig, Germany) equipped with an ESI source in negative ion mode. MS analysis was carried out in electrospray ionization (ESI) mode using negative and positive polarity for all samples with data independent MSE acquisition. The optimized parameters were obtained as follows: ionization source temperature: 70 °C, gas flow: 4 L/min, nebulizer gas (atomizer): 7.3 psi, capillary voltage: 4500 V, end plate bend voltage: 1500 V, fragmentary voltage: 280 V, and collision energy: 60 eV. An ion trap was used in the scan range *m*/*z* 100–1700 for MS and MS/MS. The chemical constituents were identified by comparing their retention index, mass spectra, and mass spectrometry fragmentation with a home-library database built by the Group of Biotechnology, Bioengineering, and Food Systems at the Far-Eastern Federal University (Russia) based on data from other spectroscopic techniques, such as nuclear magnetic resonance, ultraviolet spectroscopy, and MS, as well as data from the literature that is continually updated and revised. The capture rate was one spectrum/s for MS and two spectrum/s for MS/MS. Data collection was controlled by Windows software for Bruker Daltoniks. All experiments were replicated thrice. A four-stage ion separation mode (MS/MS mode) was implemented.

### 2.6. Statistical Analysis

Statistix v 8.1 (https://statistix.informer.com/8.1/ (accessed on 15 February 2021)) was used to compare the global yield extract (mg/100) of the three Rosa varieties according to three-way ANOVA in a completely randomized design. To more clearly present the similarities and differences of bioactive substances identified in different variants of *R. acicularis*, *R. amblyotis*, and *R. rugosa*, the team of authors used the Jaccard index. The Jaccard index, also known as the Jaccard similarity coefficient, is a statistical measure used to evaluate the similarity and diversity of sets of samples. Nine replicate samples were analyzed. A mathematical application, available at https://molbiotools.com/listcompare.php (accessed on 12 September 2024), was used to calculate the Jaccard index in this article. Thee Venn diagram was prepared using an online tool InteractiVenn (https://www.interactivenn.net/ (accessed on 24 August 2023)).

## 3. Results

### 3.1. SC-CO_2_ Extraction of Berries (R. acicularis, R. amblyotis, and R. rugosa)

Three *Rosa* varieties, i.e., *R. acicularis*, *R. amblyotis*, and *R. rugosa*, were examined by SC-CO_2_ extraction under different extraction conditions. The supercritical pressures applied ranged from 100 to 400 bar, and the extraction temperature ranged from 31 to 70 °C. The co-solvent EtOH was used in an amount of 2% of the total solvent amount. Table 1 shows the global yield of bioactive compounds by SC-CO_2_ extraction in three Rosa species. Overall, we observed that temperature and pressure in different Rosa varieties can affect the global yield.

The maximum global yield of bioactive substances from berries of *R. acicularis* was observed under the following extraction conditions:

With a pressure of 200 Bar, extraction temperature of 55 °C, and extraction time of 1 h, the global yield of biologically active substances was 6.2 mg/100 mg of plant sample; the share of the EtOH modifier was 2%.

In the case of *R. amblyotis*, it was observed that the maximum global yield of bioactive substances was observed under the following extraction conditions:

With a pressure of 250 Bar, extraction temperature of 60 °C, and extraction time of 1 h, the global yield of biologically active substances was 6.4 mg/100 mg of plant sample; the share of the EtOH modifier was 2%.

It was observed that the maximum global yield of bioactive substances from berries (*R. rugosa*) was observed under the following extraction conditions:

With a pressure of 200 Bar, extraction temperature of 60 °C, and extraction time of 1 h, the global yield of biologically active substances was 6.7 mg/100 mg of plant sample; the share of the EtOH modifier was 2%.

### 3.2. Global Metabolome Profile of R. acicularis, R. amblyotis, and R. rugosa

The structural identification of each compound was carried out on the basis of its accurate mass and MS/MS fragmentation by HPLC-ESI-ion trap-MS/MS. We were able to identify 283 chemical compounds from the extracts of *R. acicularis*, *R. amblyotis*, and *R. rugosa*: 217 chemical compounds from the polyphenol group and 66 chemical compounds from other chemical groups. The other groups included amino acids, benzaldehydes, carboxylic acids, terpenoids (sesquiterpenoids and triterpenoids), sterols, iridoids, oxylipins, fatty acids, etc. (Figure 2A). Of the 283 compounds, 62 were commonly detected from all three Rosa species. Of the three species, *R. acicularis* was the richest in the diversity of the detected compounds with 84 uniquely detected compounds followed by *R. amblyotis* (29) and *R. rugosa* (24) (Figure 2B). All identified polyphenols and other compounds along with molecular formulae and MS/MS data for *R. acicularis*, *R. amblyotis*, and *R. rugosa* are summarized in Appendix A, Table A1. The polyphenols detected in our study were categorized as flavones, flavonols, flavan-3-ols, anthocyanidins, phenolic acids, chalcones, lignans, stilbenes, coumarins, etc. In total, the metabolites detected in our study belonged to 18 compound classes. The highest number of metabolites were flavonols (71), followed by flavones (38), flavan-3-ols (13), proanthocyanidins (5), anthocyanins (24), and phenolic acids (32). These numbers indicate that SC-CO_2_ extracts of *R. acicularis*, *R. amblyotis*, and *R. rugosa* are rich in flavonoids. The highest number of chemical compounds from other groups are oxylipins (6) and terpenoids (7).

A Venn diagram showing the similarities and differences in the presence of various polyphenols in *Rosa* varieties (*R. acicularis*, *R. amblyotis*, and *R. rugosa*) is shown in Figure 2C. Forty compounds were commonly detected from the three *Rosa* varieties. These forty compounds belong to compound classes such as flavones, flavonols, anthocyanins, phenolic acids, etc., as major active compounds in SC-CO_2_ extracts of *R. acicularis*, *R. amblyotis*, and *R. rugosa*. The applied methods were able to detect 188 (*R. acicularis*), 97 (*R. amblyotis*), and 95 (*R. rugosa*) polyphenolic compounds.

Moreover, to present the similarities and differences in bioactive substances in different variations of Rosa, we used the Jaccard index (Table 2). The Jaccard index, also known as the Jaccard similarity coefficient, is a statistic used to evaluate the similarity and diversity of sets of samples [14,15]. It showed that the highest degree of similarity is present between the varieties *R. acicularis* and *R. amblyotis*—0.3689.

Twenty-nine compounds classified as anthocyanins (24) and pro-anthocyanins (five) were detected from the studied Rosa species. Generally, *R. acicularis* was rich in anthocyanins; except pelargonidins, other classes were detected. Next, *R. amblyotis* also exhibited cyanidins, delphinidins, and pelargonidins but not peonidin. *Rosa rugosa* had only seven anthocyanins belonging to cyanidins, delphinidins, and peonidin. However, it was deficient in pro-anthocyanidins (Figure 2D; Appendix A, Table A1).

To present the similarities and differences in bioactive substances in different variations of Rosa, we used the Jaccard index (Table 3). It showed that the highest degree of similarity is present between the varieties *R. acicularis* and *R. amblyotis*—0.4483.

#### 3.2.1. Hydroxy(iso)flavones

The 7-hydroxyisoflavone formononetin (compound **1**) has been already characterized as a component of *Dracocephalum jacutense* [16], *Medicago varia* [17], *Maackia amurense* [18], and Chinese herbal formula Jian-Pi-Yi-Shen pill [19]. Thus, our results indicate that the flavone formononetin is tentatively identified in the SC-CO_2_ extracts of both *R. acicularis* and *R. rugosa* (Appendix A, Table A1). The CID spectrum (collision-induced spectrum) in the positive ion mode for the flavone formononetin from the *R. rugosa* extract is shown in Figure 3. [M + H]^+^ ion produced three fragment ions with *m*/*z* 251.28, *m*/*z* 212.20, and *m*/*z* 179.93 (Figure 3). The fragment ion with *m*/*z* 251.28 produced three characteristic daughter ions with *m*/*z* 235.25, *m*/*z* 223.17, and *m*/*z* 179.08.

#### 3.2.2. Dihydroxyflavones

The flavones acacetin (compound **3**), cirsimaritin (compound **10**), cirsilineol (compound **17**), dihydroxy-dimethoxy(iso)flavone (compound **11**), and nevadensin (compound **12**) (Appendix A, Table A1) have been already characterized as a component of *Triticum aestivum* [20], *Mentha* [21], *Mexican lupine* species [22], *Artemisia annua* [23], *Rosmarinus officinalis* [24], and *Medicago varia* [17]. We also tentatively identified these flavones in SC-CO_2_ extracts of *R. acicularis*, *R. amblyotis*, and *R. rugosa*. The CID spectrum in positive ion modes for nevadensin from *R. acicularis* extracts is shown in Figure 4. The [M + H]^+^ ion produced three fragment ions with *m*/*z* 312.19, *m*/*z* 270.71, and *m*/*z* 179.05 (Figure 4). The fragment ion with *m*/*z* 312.19 produced two characteristic daughter ions with *m*/*z* 284.19 and *m*/*z* 269.13.

#### 3.2.3. Trihydroxyflavones

The flavones apigenin (compound **2**), jaceosidin (compound **14**), 5,6,4′-trihydroxy-7,8-dimethoxyflavone (compound **16**), cirsiliol (compound **17**), chrysoeriol [chrysoeriol] (compound **8**), luteolin 7-*O*-glucoside (compound **25**), kaempferide (compound **41**), etc. (Appendix A, Table A1), have already been characterized as a component of *Inula gaveolens* [25], *Lonicera henryi* [26], *Ribes meyeri* [27], and *Andean blueberry* [28]. Trihydroxyflavones were tentatively identified in SC-CO_2_ extracts of *R. acicularis*, *R. amblyotis*, and *R. rugosa*. The CID spectrum in positive ion modes for luteolin 7-*O*-glucoside from SC-CO_2_ extracts of *R. amblyotis* is shown in Figure 5. The [M + H]^+^ ion produced one fragment ion with *m*/*z* 287.19 (Figure 5). The luteolin 7-*O*-glucoside was identified in the bibliography in extracts from *Vaccinium macrocarpon* [29] and *Lonicera henryi* [26].

#### 3.2.4. Anthocyanins

Delphinidin *O*-pentoside (compound **149**), cyanidin-3-*O*-glucoside (compound **151**), delphinidin 3-*O*-glucoside (compound **154**), delphinidin 3-*O*-hexuronide (compound **157**), cyanidin 3-*O*-coumaroyl hexoside (compound **161**), etc. (Appendix A, Table A1), have already been characterized as a component of *Ribes magellanicum*; *Berberis ilicifolia*; *Berberis empetrifolia*; *Ribes maellanicum*; *Ribes cucullatum*; and *Myrteola nummalaria*. Their anthocyanins were tentatively identified in SC-CO_2_ extracts of *R. acicularis*, *R. amblyotis*, and *R. rugosa*. The CID spectrum in positive ion modes for delphinidin *O*-pentoside from SC-CO_2_ extracts of *R. rugosa* is shown in Figure 6. The [M + H]^+^ ion produced one fragment ion with *m*/*z* 303.17 (Figure 6). The fragment ion with *m*/*z* 303.17 produced three characteristic daughter ions with *m*/*z* 285.17, *m*/*z* 229.17, and *m*/*z* 165.11. The anthocyanin delphinidin *O*-pentoside was tentatively identified in the bibliography in extracts from *Myrtle* [30], *Gaultheria mucronate*, *Gaultheria antarctica* [31], and *Andean blueberry* [28].

### 3.3. Newly Detected Chemical Compounds in SC-CO_2_ Extracts of R. acicularis, R. amblyotis, and R. rugosa

Of the detected metabolites in SC-CO_2_ extracts of *R. acicularis*, *R. amblyotis*, and *R. rugosa*, forty-eight chemical constituents from the polyphenol group (15 flavones, 14 flavonols, 4 flavan-3-ols, 3 flavanones, 1 phenylpropanoid, 2 gallotannins, 1 ellagitannin, 4 phenolic acids, 1 dihydrochalcone, and 3 coumarins) were identified for the first time. The newly identified polyphenols include flavones (acacetin, calycosin, diosmetin, odoratin, salvigenin, jaceosidin, nevadensin, cirsilineol, scutellarin, chrysoeriol 6-O-hexoside, chrysoeriol 6-*O*-glucoside, chrysoeriol 8-*O*-glucoside, chrysin 6-*C*-[2″-*O*-glucoside] -glucoside, and chrysin di-*O*-glucoside, lonicerin), flavonols (rhamnetin I, rhamnetin II, taxifolin 3-*O*-arabinofuranoside, isorhamnetin-3-*O*-arabinoside, isorhamnetin 3-*O*-pentoside, astragalin, dihydrokaempferol-O-hexoside, aromadendrin O-hexoside, taxifolin-3-*O*-glucoside, taxifolin-3-*O*-hexoside, quercetin 7-*O*-glucoside, quercetin-hexuronide, quercetin *O*-hexoside *O*-deoxyhexoside, and bioquercetin), flavan-3-ols (afzelechin, (*epi*)-afzelechin, gallocatechin, and (*epi*)-gallocatechin), flavanones (eriodictyol, hesperitin, and taxifolin-3-*O*-rhamnoside), phenylpropanoid vimalin, etc. (Appendix A, Table A1).

## 4. Discussion

Since the most biologically active substances in plants are usually polyphenolic compounds, extraction is the first and most important step in the stepwise separation of these compounds [32]. The most common extractions applicable to phenolic compounds are liquid/liquid and solid/liquid, and they are often used specifically for the separation of these components. The phenolic structure of polyphenols makes them relatively hydrophilic, so free phenolic compounds, including aglycones, glycosides, and oligomers are extracted using water, polar organic solvents such as ethyl acetate, methanol, ethanol, acetonitrile, acetone, and their mixtures with water or non-polar solvents (chloroform and diethyl ether) [4]. Nowadays, as a result of the negative impact of industrial extraction on the environment, the concept of green extraction has been introduced to protect both the environment and consumers. This also directly affects the increasing competition of industries to be more environmentally friendly (use of by-products and biodegradability), economical (lower energy and solvent consumption), and innovative [33]. In line with this approach to green extraction, non-traditional extraction methods such as microwave, ultrasonic extraction, and methods based on the use of compressed fluids as extracting agents such as subcritical water extraction (SWE), supercritical fluid extraction (SFE), pressurized fluid extraction (PFE), or accelerated solvent extraction (ASE) are used for the actual separation of phenolic compounds [34,35]. Generally, the most common extraction methods for phenolic compounds use acetone [36,37] or methanol [38] as phenolic compound extracting agents, but from the point of view of use in food and cosmetic industries, ethanol and water are definitely preferred [39].

As shown earlier, bioactive compounds of aboveground plant parts are efficiently extracted using organic solvents such as methanol and ethanol. However, the extraction products at the final stage require additional costs in terms of purification from trace amounts of the solvents used. SC-CO_2_ extraction, which is rightfully classified as a “green” extraction method, can be used as an alternative to traditional extraction methods: maceration or Soxhlet extraction [40,41]. SC-CO_2_ extraction has been used in the evaluation of food products, the isolation of bioactive substances, and the determination of lipid levels in food products and the levels of toxic substances. With SC-CO_2_ extraction, the products do not contain residues of organic solvents that are encountered in conventional extraction methods, and the solvents can be toxic, for example in the case of methanol and n-hexane. Easy removal of solvent from the final product, high selectivity, and moderate extraction temperatures are the main attractive features of SC-CO_2_ technology, which leads to a significant increase in research for applications in cosmetic, pharmaceutical, and food industries. When comparing possible supercritical solvents, carbon dioxide has the most attractive advantages: non-toxic, non-flammable, environmentally friendly, and, most importantly, a renewable resource [42]. Previously, our group of authors successfully used supercritical CO_2_ extraction in the extraction of the polyphenolic composition of the forage grass *Medicago varia* [17], the extraction of the aromatic component from *Ribes fragrans* leaves [43], the refinement of the polyphenolic composition of medicinal plants *Ledum palustre* [44] and *Maackia amurense* [18], the isolation of the sulfate compounds from *Zostera marina* [45], etc. Thus, the use of SC-CO_2_ extraction is an effective approach for the extraction of bioactive compounds.

High oxidation–reduction potential is the most important property of flavonoids, which allows them to act as singlet hydrogen quenchers, reducing agents, and hydrogen donors. Flavonoids have metal chelating potential [46]. Regular consumption of flavonoids has been directly associated with a reduced incidence of various cancers and heart disease [47]. There is currently great interest in flavonoid research due to the possibility of direct effects on health through diet, in which case preventive health care can be achieved through the consumption of fruits and vegetables. Flavonols are a class of flavonoids commonly found in many fruits and vegetables, and their content varies greatly depending on environmental factors such as growing conditions, climate, storage, and cooking conditions [48].

A total of 75 flavonols have been identified in SC-CO_2_ extracts of *R. acicularis*, *R. amblyotis*, and *R. rugosa*, of which 14 compounds (rhamnetin I, rhamnetin II, taxifolin 3-*O*-arabinofuranoside, isorhamnetin-3-*O*-arabinoside, isorhamnetin 3-*O*-pentoside, astragalin, dihydrokaempferol-*O*-hexoside, aromadendrin *O*-hexoside, taxifolin-3-*O*-glucoside, taxifolin-3-*O*-hexoside, quercetin 7-*O*-glucoside, quercetin-hexuronide, quercetin *O*-hexoside *O*-deoxyhexoside, and bioquercetin) were identified for the first time in *Rose* SC-CO_2_ extracts, highlighting the great potential for applications in functional food technologies.

Flavanones can be characterized by the presence of a saturated three-carbon chain and an oxygen atom at the C4 position. They are usually glycosylated with a disaccharide at the C7 position. Flavanones are present in high concentrations only in citrus fruits but have also been identified in tomatoes and some aromatic plants (mint). The main aglycones are naringenin in grapefruit, hesperetin in oranges, and eriodictyol in lemons. A total of five flavanones have been identified in berry extracts, of which three compounds (hesperetin, quercetin *O*-hexoside *O*-deoxyhexoside, and taxifolin-3-*O*-rhamnoside) were identified for the first time in SC-CO_2_ extracts of *R. acicularis*, *R. amblyotis*, and *R. rugosa*.

Anthocyanins are water-soluble pigments that turn red, purple, or blue depending on pH. They are also flavonoids but are synthesized via the phenylpropanoid pathway. Anthocyanins are found in all plant matrices, including leaves, stems, and roots, with particularly high concentrations in flowers and fruits. Anthocyanidins are the main structures of anthocyanins. Anthocyanidins consist of an aromatic ring A linked to a heterocyclic ring C containing oxygen, which is also linked via a carbon–carbon bond to a third aromatic ring B [49]. Many researchers have noted that glycoside derivatives of three unmethylated anthocyanidins (pelargonidin, cyanidin, and delphinidin) are the most common in nature, occurring in 69% of fruits, 50% of flowers, and 80% of pigmented leaves [50]. Six anthocyanidins are most commonly found in plants: delphinidin, pelargonidin, malvidin, cyanidin, peonidin, petunidin, and malvidin. Many anthocyanins have sugar residues acylated with aromatic or aliphatic acids [51]. In our studies, we were able to identify anthocyanins belonging to the following groups: delphinidin, pelargonidin, cyanidin, and peonidin—a total of 24 compounds from SC-CO_2_-extracts of *R. acicularis*, *R. amblyotis*, and *R. rugosa*.

## 5. Conclusions

*Rosa* species contain many polyphenolic components and components of other chemical groups that have valuable biological activities. SC-CO_2_ extraction of *R. acicularis*, *R. amblyotis*, and *R. rugosa* was successfully carried out by the team of authors; certain extraction conditions were selected. The extracts obtained showed both a high content of polyphenolic composition and a high content of the saponin group. Tandem mass spectrometry (HPLC-ESI—ion trap) was used to detect target analytes. Mass spectrometric data were recorded on an ion trap equipped with an ESI source in negative and positive ion mode. A four-stage ion separation mode is implemented. Two hundred and eighty-seven different biologically active compounds were found in SC-CO_2_ extracts of *R. acicularis*, *R. amblyotis*, and *R. rugosa*. For the first time, forty-eight chemical constituents from the polyphenol group (15 flavones, 14 flavonols, 4 flavan-3-ols, 3 flavanones, 1 phenylpropanoid, 2 gallotannins, 1 ellagitannin, 4 phenolic acids, 1 dihydrochalcone, and 3 coumarins) were identified in supercritical extracts of *Rosa acicularis*, *Rosa amblyotis*, and *Rosa rugosa*. Our research has shown for the first time the acceptability and effectiveness of the supercritical extraction method in extracting the largest number of chemical compounds of both the polyphenol group and compounds of other groups, which can be a start in improving applied food and pharmaceutical technologies.

## Figures and Tables

**Figure 1 plants-14-00059-f001:**
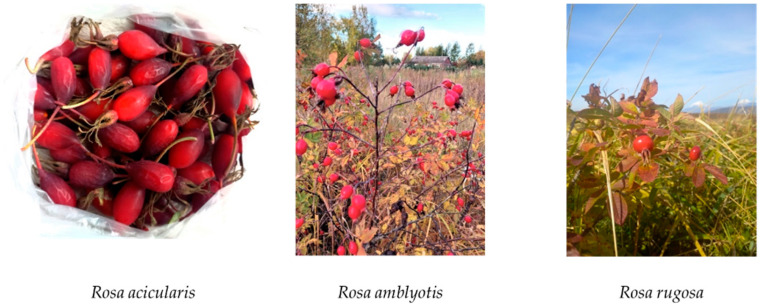
Pictures of the studied three Rosa species.

**Figure 2 plants-14-00059-f002:**
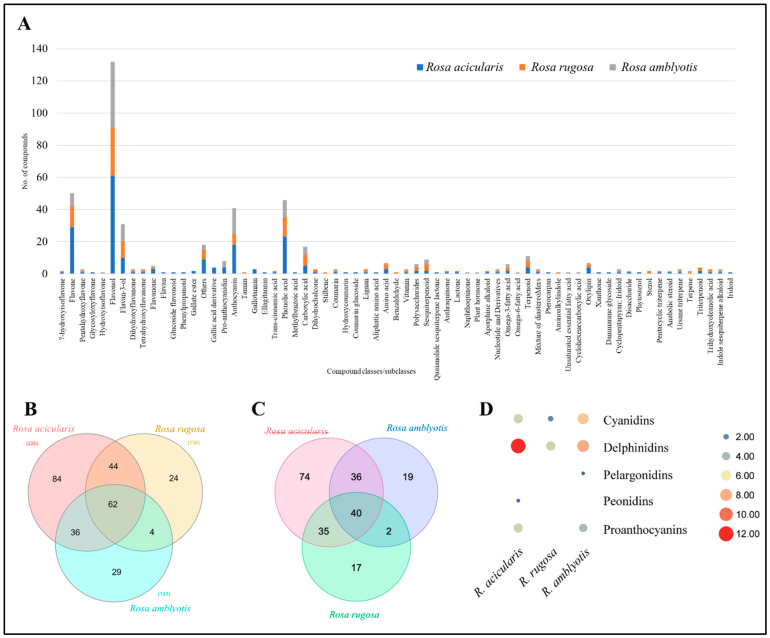
Global metabolome profile of three Rosa species. (**A**) Bar plot showing compound classes (and subclasses) and respective number of compounds detected in each Rosa species. (**B**) Venn diagram showing number of common and specific compounds detected in three Rosa species. (**C**) Venn diagram showing number of common and specific polyphenolic compounds detected in three Rosa species. (**D**) Scatter plot showing number of anthocyanins detected in each Rosa species.

**Figure 3 plants-14-00059-f003:**
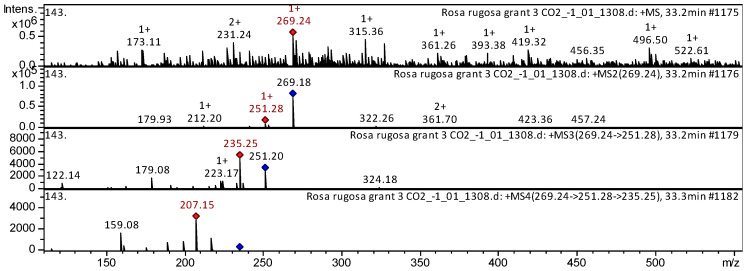
CID spectrum of formononetin from SC-CO_2_ extracts of *R. rugosa*, *m*/*z* 269.24. Above is an MS scan in the range 100–1700 *m*/*z* and below is fragmentation spectra (top to bottom): MS2 of protonated formononetin ion (269.24 *m*/*z*, red diamond), MS3 fragment 269.24 → 251.28 *m*/*z*, and MS4 fragment 269.24 → 251.28 → 235.25 *m*/*z*.

**Figure 4 plants-14-00059-f004:**
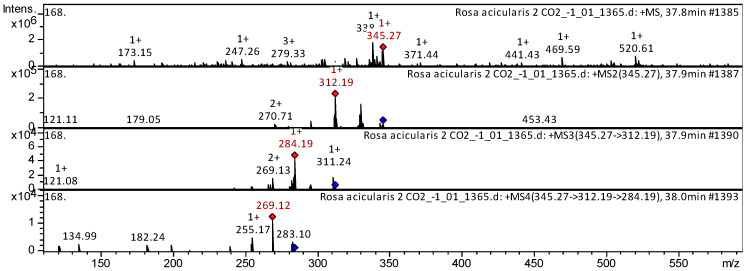
CID spectrum of nevadensin from SC-CO_2_ extracts of *R. acicularis*, *m*/*z* 345.27. Above is an MS scan in the range 100–1700 *m*/*z* and below is fragmentation spectra (top to bottom): MS2 of protonated nevadensin ion (345.27 *m*/*z*, red diamond), MS3 fragment 345.27 → 312.19 *m*/*z*, and MS4 fragment 345.27 → 312.19 → 284.19 *m*/*z*.

**Figure 5 plants-14-00059-f005:**
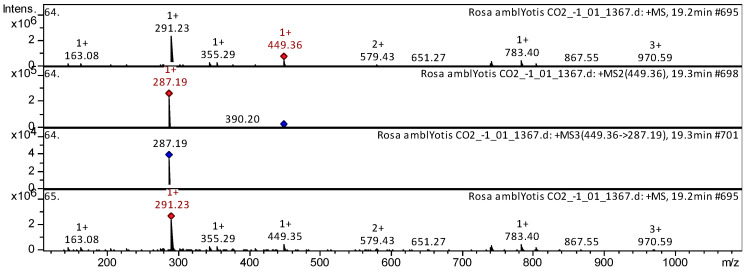
CID spectrum of luteolin-7-*O*-glucoside from SC-CO_2_ extracts of *R. amblyotis*, *m*/*z* 449.36. Above is an MS scan in the range 100–1700 *m*/*z* and below is fragmentation spectra (top to bottom): MS2 of protonated luteolin-7-*O*-glucoside ion (449.36 *m*/*z*, red diamond), MS3 fragment 449.36 → 287.19 *m*/*z*, and MS4 fragment 449.36 → 287.19 → 287.19 *m*/*z*.

**Figure 6 plants-14-00059-f006:**
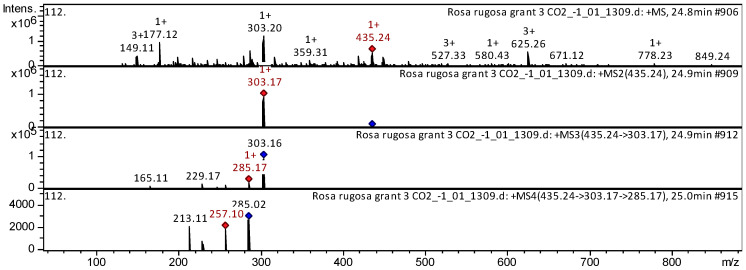
CID spectrum of delphinidin *O*-pentoside from SC-CO_2_ extracts of *R. rugosas*, *m*/*z* 435.24. Above is an MS scan in the range 100–1700 *m*/*z* and below is fragmentation spectra (top to bottom): MS2 of protonated delphinidin *O*-pentoside ion (435.24 *m*/*z*, red diamond), MS3 fragment 435.24 → 303.17 *m*/*z*, and MS4 fragment 435.24 → 303.17 → 285.17 *m*/*z*.

**Table 1 plants-14-00059-t001:** The global yield of extract (mg/100 mg) after SC-CO_2_ extraction of berries of three Rosa species.

Species	Temperature °C /Pressure (Bar)	50	100	150	200	250	300	350
*R. acicularis*	31	0.29 ± 0.012 u	1.23 ± 0.058 r	1.86 ± 0.058 pq	2.03 ± 0.058 k–q	2.30 ± 0.01 f–l	2.66 ± 0.115 Za–d	1.96 ± 0.057 m–q
*R. amblyotis*		0.29 ± 0.01 u	1.16 ± 0.058 r	1.83 ± 0.11 q	1.93 n–q	2.26 ± 0.058 f–m	2.56 ± 0.058 a–f	2.23 ± 0.058 g–n
*R. rugosa*		0.26 ± 0.058 u	1.23 ± 0.058 r	1.93 ± 0.058 n–q	2.10 ± 0.1 j–q	2.33 ± 0.058 e–k	2.63 ± 0.058 Za–e	2.26 ± 0.058 f–m
*R. acicularis*	40	0.43 ± 0.059 Tu	1.96 ± 0.058 m–q	2.13 ± 0.055 i–q	2.36 ± 0.1 d–j	2.73 ± 0.05 X–Za–c	2.93± 0.052 T–Z	20 ± 0.058 l–q
*R. amblyotis*		0.76 ± 0.25 s	1.83 ± 0.058 q	2.16 ± 0.058 h–p	2.36 ± 0.058 d–j	2.70 ± 0.0 Yza–c	2.90 ± 0.0 U–Z	2.50 ± 0.0 b–g
*R. rugosa*		1.30 ± 0.3 r	1.90 ± 0.0 o–q	2.10 ± 0.0 j–q	2.30 ± 0.0 f–l	2.70 ± 0.0 Yza–c	2.90 ± 0.0 U–Z	2.50 ± 0.0 b–g
*R. acicularis*	45	0.70± 0.0 st	2.70 ± 0.1 Yza–c	2.56 ± 0.057 a–f	3.30 ± 0.1 RS	3.06 ± 0.0577 S–W	3.06 ± 0.058 S–W	30 ± 0.1 S–Y
*R. amblyotis*		0.70 ± 0.0 st	2.70 ± 0.0 Yza–c	2.50 ± 0.0 b–g	3.30 ± 0.0 RS	30 ± 0.0 S–Y	3.06 ± 0.058 S–W	3.03 ± 0.058 S–X
*R. rugosa*		1.43 ± 0.252 r	2.70 ± 0.0 Yza–c	2.50 ± 0.0 b–g	3.30 ± 0.0 RS	30 ± 0.0 S–Y	3.10 ± 0.0 S–V	3.13 ± 0.058 S–V
*R. acicularis*	50	2.50 ± 0.0 b–g	3.20 ± 0.0 RSTU	4.03 ± 0.058 M–O	50 ± 0.1 FG	3.96 ± 0.57 N–P	30 ± 0.1 S–Y	3.06 ± 0.11 S–W
*R. amblyotis*		2.56 ± 0.058 a–f	3.26 ± 0.058 RS	4.30 ± 0.10 J–M	4.56 ± 0.058 IJ	4.46 ± 0.058 I–K	3.13 ± 0.115 S–V	3.20 ± 0.0 R–U
*R. rugosa*		2.20 ± 0.265 g–o	4.06 ± 0.115 L–O	4.30 ± 0.0 J–M	4.50 ± 0.0 IJ	4.50 ± 0.0 IJ	3.50 ± 0.0 QR	4.10 ± 0.0 L–O
*R. acicularis*	55	2.66 ± 0.05 Za–d	3.70 ± 0.10 PQ	4.43 ± 0.11 I–K	6.16 ± 0.05 B–D	5.20 ± 0.10 E–G	5.20 ± 0.0 E–G	3.96 ± 0.059 N–P
*R. amblyotis*		2.56 ± 0.058 a–f	3.70 ± 0.0 PQ	4.56 ± 0.058 IJ	5.46 ± 0.058 E	5.23 ± 0.058 E–G	5.20 ± 0.0 E–G	3.83 ± 0.115 OP
*R. rugosa*		2.46 ± 0.231 b–h	3.70 ± 0.0 PQ	4.36 ± 0.115 I–L	5.50 ± 0.0 E	5.20 ± 0.0 E–G	5.20 ± 0.0 E–G	5.10 ± 0.0 FG
*R. acicularis*	60	2.43 ± 0.12 c–i	3.10 ± 0.0 S–V	3.93 ± 0.057 N–P	4.50 ± 0.10 IJ	50 ± 0.0 FG	4.36 ± 0.057 I–L	3.50 ± 0.10 QR
*R. amblyotis*		2.43 ± 0.058 c–i	3.16 ± 0.058 S–U	5.06 ± 0.057 FG	5.30 ± 0.0 EF	6.33 ± 0.06 BC	60 ± 0.0 D	3.50 ± 0.0 QR
*R. rugosa*		2.46 ± 0.058 b–h	3.10 ± 0.0 S–V	60 ± 0.0 D	6.7 ± 0.0 A	6.46 ± 0.058 AB	5.93 ± 0.058 D	4.96 ± 0.231 GH
*R. acicularis*	70	2.30 ± 0.9 f–l	2.83 ± 0.058 V–Za	3.83 ± 0.058 OP	3.90 ± 0.10 N–P	3.06 ± 0.11 S–W	3.23 ± 0.057 R–T	2.93 ± 0.057 T–Z
*R. amblyotis*		2.66 ± 0.153 Za–d	2.83 ± 0.115 V–Za	3.86 ± 0.058 N–P	3.93 ± 0.153 N–P	5.03 ± 0.153 FG	3.93 ± 0.058 N–P	3.23 ± 0.252 R–T
*R. rugosa*		2.76 ± 0.058 W–Zab	2.90 ± 0.1 U–Z	40 ± 0.1 M–P	4.93 ± 0.153 GH	6.10 ± 0.1 CD	4.66 ± 0.058 HI	4.16 ± 0.058 K–N

Note: The values presented are means (n = 3) ± standard deviation. Different letters indicate significant differences among the mean values.

**Table 2 plants-14-00059-t002:** Jaccard Index for three varieties of wild *Rosa*.

	*Rosa acicularis* (185)	*Rosa amblyotis* (97)	*Rosa rugosa* (94)
*Rosa acicularis* (185)	--	76 0.3689	75 0.3676
*Rosa amblyotis* (97)	76 0.3689	--	42 0.2819
*Rosa rugosa* (94)	75 0.3676	42 0.2819	--

**Table 3 plants-14-00059-t003:** Jaccard index for three varieties of wild *Rosa*.

	*Rosa acicularis* (22)	*Rosa amblyotis* (20)	*Rosa rugosa* (7)
*Rosa acicularis* (22)	--	13 0.4483	7 0.3182
*Rosa amblyotis* (20)	13 0.4483	--	4 0.1739
*Rosa rugosa* (7)	7 0.3182	4 0.1739	--

## Data Availability

Data are contained within the article.

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
