# Peer review of "Application of Supercritical CO_2_ Extraction and Identification of Polyphenolic Compounds in Three Species of Wild Rose from Kamchatka: *Rosa acicularis*, *Rosa amblyotis*, and *Rosa rugosa"

_plants, 2024, doi:10.3390/plants14010059_

Round 1

Reviewer 1 Report

Comments and Suggestions for Authors

Why the authors did not chose a factorial design to explore the factors role in  the global yields? The graphics would be much better than the ones presented. The authors could also obtain an equation of the pressure and temperature role to predict yields.

Global yield graphics should be much improved. 

Author Response

Authors responses to reviewer 1

Reviewer 1: Why the authors did not choose a factorial design to explore the factors’ role in the global yields? The graphics would be much better than the ones presented. The authors could also obtain an equation of the pressure and temperature role to predict yields.

Our response: Respected reviewer 1, thank you for your valuable comments and time spent on evaluation of our submission. We have now revised the submission in light of comments. All the changes are presented using Track-changes function.

We have now used a three-way factorial design and Tukey’s honestly significant difference test and presented the yield results in form of a table. Please see the revised manuscript and Table 1.

Reviewer 1: Global yield graphics should be much improved. 

Our response: Respected reviewer 1, since global yield data is now collectively given in the Table 1, hence we have opted to remove the figures related to this data to avoid redundancy.

Reviewer 2 Report

Comments and Suggestions for Authors

Comments to the Authors:
The authors of this paper present an interesting comparative metabolomic study of three varieties of wild Rosa via extraction with SC-CO2. However, some details should be considered by the authors:

COMMENT: Page 2, line 51: More (recent) references could be added.

COMMENT: Page 2, lines 58-59: More (recent) references could be added.

COMMENT: Page 2, line 60-61: More (recent) references could be added.

COMMENT: Page 2, line 64, “… and cancer (Gordon, 1996): A reference should be added.

COMMENT: Page 2, line 67: More (recent) references could be added.

COMMENT: Page 2, lines 68-70: Some references could be added.

COMMENT: Page 4, line 130, “All experiments were replicated thrice.: A discussion about possible experimental errors-deviations could be added.

COMMENT: Page 5, lines 152-156, Page 6, lines 165-170, Page 7, lines 179-184,: Some comments about the maximum global yield and the extraction conditions could be added.

COMMENT: Page 13, lines 335-340: More comments on the results of these previously published studies could also be added to this article.

The experimental results support the conclusions of the authors and I think that this paper may be published.

Author Response

Authors responses to reviewer 2

The authors of this paper present an interesting comparative metabolomic study of three varieties of wild Rosa via extraction with SC-CO2. However, some details should be considered by the authors:

The experimental results support the conclusions of the authors and I think that this paper may be published.

Our response: Respected reviewer 2, thank you for your valuable comments and time spent on evaluation of our submission. We have now revised the submission in light of comments. All the changes are presented using Track-changes function.

Reviewer 2: Page 2, line 51: More (recent) references could be added.

Reviewer 2:  Page 2, lines 58-59: More (recent) references could be added.

Reviewer 2: Page 2, line 60-61: More (recent) references could be added.

Reviewer 2: Page 2, line 64, “… and cancer (Gordon, 1996)”: A reference should be added.

Reviewer 2: Page 2, line 67: More (recent) references could be added.

Reviewer 2: Page 2, lines 68-70: Some references could be added.

Reviewer 2: Page 4, line 130, “All experiments were replicated thrice.”: A discussion about possible experimental errors-deviations could be added.

Reviewer 2: Page 5, lines 152-156, Page 6, lines 165-170, Page 7, lines 179-184: Some comments about the maximum global yield and the extraction conditions could be added.

Our response: Dear reviewer 2, we have now used a three-way factorial design and Tukey’s honestly significant difference test and presented the yield results in form of a table. Please see the revised manuscript and Table 1. Hence, we believe that this test indicative has significantly improved the overall presentation of the data. Relevant statements have been updated.

Reviewer 2: Page 13, lines 335-340: More comments on the results of these previously published studies could also be added to this article.

Reviewer 3 Report

Comments and Suggestions for Authors

Dear authors,

please find attached the complete review report with my comments and suggestions.

Best regards

Author Response

Dear Reviewer.
Thank you very much for your invaluable time spent reviewing our manuscript. We greatly appreciate your comments.
We have significantly revised the manuscript based on your comments, adding both statistical analyses and reformatting the graphs.
We kindly ask you to review the manuscript again.

Round 2

Reviewer 1 Report

Comments and Suggestions for Authors

Suggest to approve

Reviewer 2 Report

Comments and Suggestions for Authors

The experimental results support the conclusions of the authors and I think that this paper may be published.